# Towards modular and programmable architecture search

**Renato Negrinho**[1] *    **Darshan Patil**[1]    **Nghia Le**[1]    **Daniel Ferreira**[2]
**Matthew R. Gormley**[1]    **Geoffrey Gordon**[1,3]
Carnegie Mellon University[1], TU Wien[2], Microsoft Research Montreal[3]

## Abstract

Neural architecture search methods are able to find high performance deep learning architectures with minimal effort from an expert [1]. However, current systems focus on specific use-cases (e.g. convolutional image classifiers and recurrent language models), making them unsuitable for general use-cases that an expert might wish to write. Hyperparameter optimization systems [2, 3, 4] are general-purpose but lack the constructs needed for easy application to architecture search. In this work, we propose a formal language for encoding search spaces over general computational graphs. The language constructs allow us to write modular, composable, and reusable search space encodings and to reason about search space design. We use our language to encode search spaces from the architecture search literature. The language allows us to decouple the implementations of the search space and the search algorithm, allowing us to expose search spaces to search algorithms through a consistent interface. Our experiments show the ease with which we can experiment with different combinations of search spaces and search algorithms without having to implement each combination from scratch. We release an implementation of our language with this paper[2].

## 1 Introduction

Architecture search has the potential to transform machine learning workflows. High performance deep learning architectures are often manually designed through a trial-and-error process that amounts to trying slight variations of known high performance architectures. Recently, architecture search techniques have shown tremendous potential by improving on handcrafted architectures, both by improving state-of-the-art performance and by finding better tradeoffs between computation and performance. Unfortunately, current systems fall short of providing strong support for general architecture search use-cases.

Hyperparameter optimization systems [2, 3, 4, 5] are not designed specifically for architecture search use-cases and therefore do not introduce constructs that allow experts to implement these use-cases efficiently, e.g., easily writing new search spaces over architectures. Using hyperparameter optimization systems for an architecture search use-case requires the expert to write the encoding for the search space over architectures as a conditional hyperparameter space and to write the mapping from hyperparameter values to the architecture to be evaluated. Hyperparameter optimization systems are completely agnostic that their hyperparameter spaces encode search spaces over architectures.

By contrast, architecture search systems [1] are in their infancy, being tied to specific use-cases (e.g., either reproducing results reported in a paper or concrete systems, e.g., for searching over Scikit-Learn pipelines [6]) and therefore lack support for general architecture search workflows. For

example, current implementations of architecture search methods rely on ad-hoc encodings for search spaces, providing limited extensibility and programmability for new work to build on. For example, implementations of the search space and search algorithm are often intertwined, requiring substantial coding effort to try new search spaces or search algorithms.

**Contributions** We describe a modular language for encoding search spaces over general computational graphs. We aim to improve the programmability, modularity, and reusability of architecture search systems. We are able to use the language constructs to encode search spaces in the literature. Furthermore, these constructs allow the expert to create new search spaces and modify existing ones in structured ways. Search spaces expressed in the language are exposed to search algorithms under a consistent interface, decoupling the implementations of search spaces and search algorithms. We showcase these functionalities by easily comparing search spaces and search algorithms from the architecture search literature. These properties will enable better architecture search research by making it easier to benchmark and reuse search algorithms and search spaces.

## 2 Related work

**Hyperparameter optimization** Algorithms for hyperparameter optimization often focus on small or simple hyperparameter spaces (e.g., closed subsets of Euclidean space in low dimensions). Hyperparameters might be categorical (e.g., choice of regularizer) or continuous (e.g., learning rate and regularization constant). Gaussian process Bayesian optimization [7] and sequential model based optimization [8] are two popular approaches. Random search has been found to be competitive for hyperparameter optimization [9, 10]. Conditional hyperparameter spaces (i.e., where some hyperparameters may be available only for specific values of other hyperparameters) have also been considered [11, 12]. Hyperparameter optimization systems (e.g. Hyperopt [2], Spearmint [3], SMAC [5, 8] and BOHB [4]) are general-purpose and domain-independent. Yet, they rely on the expert to distill the problem into an hyperparameter space and write the mapping from hyperparameter values to implementations.

**Architecture search** Contributions to architecture search often come in the form of search algorithms, evaluation strategies, and search spaces. Researchers have considered a variety of search algorithms, including reinforcement learning [13], evolutionary algorithms [14, 15], MCTS [16], SMBO [16, 17], and Bayesian optimization [18]. Most search spaces have been proposed for recurrent or convolutional architectures [13, 14, 15] focusing on image classification (CIFAR-10) and language modeling (PTB). Architecture search encodes much of the architecture design in the search space (e.g., the connectivity structure of the computational graph, how many operations to use, their type, and values for specifying each operation chosen). However, the literature has yet to provide a consistent method for designing and encoding such search spaces. Systems such as Auto-Sklearn [19], TPOT [20], and Auto-Keras [21] have been developed for specific use-cases (e.g., Auto-Sklearn and TPOT focus on classification and regression of featurized vector data, Auto-Keras focus on image classification) and therefore support relatively rigid workflows. The lack of focus on extensibility and programmability makes these systems unsuitable as frameworks for general architecture search research.

## 3 Proposed approach: modular and programmable search spaces

To maximize the impact of architecture search research, it is *fundamental* to improve the programmability of architecture search tools[3]. We move towards this goal by designing *a language to write search spaces over computational graphs*. We identify the following advantages for our language and search spaces encoded in it:

- **Similarity to computational graphs:** Writing a search space in our language is similar to writing a fixed computational graph in an existing deep learning framework. The main difference is that nodes in the graph may be search spaces rather than fixed operations (e.g., see Figure 5). A search space maps to a single computational graph once all its hyperparameters have been assigned values (e.g., in frame d in Figure 5).

- **Modularity and reusability:** The building blocks of our search spaces are modules and hyperparameters. Search spaces are created through the composition of modules and their interactions. Implementing a new module only requires dealing with aspects local to the module. Modules and hyperparameters can be reused across search spaces, and new search spaces can be written by combining existing search spaces. Furthermore, our language supports search spaces in general domains (e.g., deep learning architectures or Scikit-Learn [22] pipelines).

- **Laziness:** A *substitution module* delays the creation of a subsearch space until all hyperparameters of the substitution module are assigned values. Experts can use substitution modules to encode natural and complex conditional constructions by concerning themselves only with the conditional branch that is chosen. This is simpler than the support for conditional hyperparameter spaces provided by hyperparameter optimization tools, e.g., in Hyperopt [2], where all conditional branches need to be written down explicitly. Our language allows conditional constructs to be expressed implicitly through composition of language constructs (e.g., nesting substitution modules). Laziness also allows us to encode search spaces that can expand infinitely, which is not possible with current hyperparameter optimization tools (see Appendix D.1).

- **Automatic compilation to runnable computational graphs:** Once all choices in the search space are made, the single architecture corresponding to the terminal search space can be mapped to a runnable computational graph (see Algorithm 4). By contrast, for general hyperparameter optimization tools this mapping has to be written manually by the expert.

# 4 Components of the search space specification language

A search space is a graph (see Figure 5) consisting of hyperparameters (either of type independent or dependent) and modules (either of type basic or substitution). This section describes our language components and show encodings of simple search spaces in our Python implementation. Figure 5 and the corresponding search space encoding in Figure 4 are used as running examples. Appendix A and Appendix B provide additional details and examples, e.g. the recurrent cell search space of [23].

**Independent hyperparameters**    The value of an independent hyperparameter is chosen from its set of possible values. An independent hyperparameter is created with a set of possible values, but without a value assigned to it. Exposing search spaces to search algorithms relies mainly on iteration over and value assignment to independent hyperparameters. An independent hyperparameter in our implementation is instantiated as, for example, `D([1, 2, 4, 8])`. In Figure 5, `IH-1` has set of possible values $\{64, 128\}$ and is eventually assigned value $64$ (shown in frame d).

**Dependent hyperparameters**    The value of a dependent hyperparameter is computed as a function of the values of the hyperparameters it depends on (see line 7 of Algorithm 1). Dependent hyperparameters are useful to encode relations between hyperparameters, e.g., in a convolutional network search space, we may want the number of filters to increase after each spatial reduction. In our implementation, a dependent hyperparameter is instantiated as, for example, `h = DependentHyperparameter(lambda dh: 2*dh["units"], {"units": h_units})`. In Figure 5, in the transition from frame `a` to frame `b`, `IH-3` is assigned value 1, triggering the value assignment of `DH-1` according to its function `fn:2*x`.

**Basic modules**    A basic module implements computation that depends on the values of its properties. Search spaces involving only basic modules and hyperparameters do not create new modules or hyperparameters, and therefore are fixed computational graphs (e.g., see frames `c` and `d` in Figure 5). Upon compilation, a basic module consumes the values of its inputs, performs computation, and publishes the results to its outputs (see Algorithm 4). Deep learning layers can be wrapped as basic modules, e.g., a

```
def one_layer_net():                              1
    a_in, a_out = dropout(D([0.25, 0.5]))         2
    b_in, b_out = dense(D([100, 200, 300]))       3
    c_in, c_out = relu()                          4
    a_out["out"].connect(b_in["in"])              5
    b_out["out"].connect(c_in["in"])              6
    return a_in, c_out                            7
```

Figure 1: Search space over feedforward networks with dropout rate of 0.25 or 0.5, ReLU activations, and one hidden layer with 100, 200, or 300 units.

fully connected layer can be wrapped as a single-input single-output basic module with one hyperparameter for the number of units. In the search space in Figure 1, `dropout`, `dense`, and `relu` are

basic modules. In Figure 5, both frames `c` and `d` are search spaces with only basic modules and hyperparameters. In the search space of frame `d`, all hyperparameters have been assigned values, and therefore the single architecture can be mapped to its implementation (e.g., in Tensorflow).

**Substitution modules** Substitution modules encode structural transformations of the computational graph that are delayed[4] until their hyperparameters are assigned values. Similarly to a basic module, a substitution module has hyperparameters, inputs, and outputs. Contrary to a basic module, a substitution module does not implement computation—it is substituted by a subsearch space (which depends on the values of its hyperparameters and may contain new substitution modules). Substitution is triggered once all its hyperparameters have been assigned

```
def multi_layer_net():                          1
    h_or = D([0, 1])                            2
    h_repeat = D([1, 2, 4])                     3
    return siso_repeat(                         4
        lambda: siso_sequential([               5
            dense(D([300])),                    6
            siso_or([relu, tanh], h_or)         7
        ]), h_repeat)                           8
```

Figure 2: Search space over feedforward networks with 1, 2, or 4 hidden layers and ReLU or tanh activations.

values. Upon substitution, the module is removed from the search space and its connections are rerouted to the corresponding inputs and outputs of the generated subsearch space (see Algorithm 1 for how substitutions are resolved). For example, in the transition from frame b to frame c of Figure 5, `IH-2` was assigned the value 1 and `Dropout-1` and `IH-7` were created by the substitution of `Optional-1`. The connections of `Optional-1` were rerouted to `Dropout-1`. If `IH-2` had been assigned the value 0, `Optional-1` would have been substituted by an identity basic module and no new hyperparameters would have been created. Figure 2 shows a search space using two substitution modules: `siso_or` chooses between `relu` and `tanh`; `siso_repeat` chooses how many layers to include. `siso_sequential` is used to avoid multiple calls to `connect` as in Figure 1.

**Auxiliary functions** Auxiliary functions, while not components per se, help create complex search spaces. Auxiliary functions might take functions that create search spaces and put them together into a larger search space. For example, the search space in Figure 3 defines an auxiliary RNN cell that captures the high-level functional dependency: $h_t = q_h(x_t, h_{t-1})$ and $y_t = q_y(h_t)$. We can instantiate a specific search space as `rnn_cell(lambda:`

```
def rnn_cell(hidden_fn, output_fn):                 1
    h_inputs, h_outputs = hidden_fn()               2
    y_inputs, y_outputs = output_fn()               3
    h_outputs["out"].connect(y_inputs["in"])        4
    return h_inputs, y_outputs                       5
```

Figure 3: Auxiliary function to create the search space for the recurrent cell given functions that create the subsearch spaces.

`siso_sequential([concat(2), one_layer_net()]), multi_layer_net)`.

## 5 Example search space

We ground discussion textually, through code examples (Figure 4), and visually (Figure 5) through an example search space. There is a convolutional layer followed, optionally, by dropout with rate 0.25 or 0.5. After the optional dropout layer, there are two parallel chains of convolutional layers. The first chain has length 1, 2, or 4, and the second chain has double the length of the first. Finally, the outputs of both chains are concatenated. Each convolutional layer has 64 or 128 filters (chosen separately). This search space has 25008 distinct models.

Figure 5 shows a sequence of graph transitions for this search space. `IH` and `DH` denote type identifiers for independent and dependent hyperparameters, respectively. Modules and hyperpa-

```
def search_space():                                          1
    h_n = D([1, 2, 4])                                       2
    h_ndep = DependentHyperparameter(                        3
        lambda dh: 2 * dh["x"], {"x": h_n})                  4
                                                             5
    c_inputs, c_outputs = conv2d(D([64, 128]))               6
    o_inputs, o_outputs = siso_optional(                     7
        lambda: dropout(D([0.25, 0.5])), D([0, 1]))          8
    fn = lambda: conv2d(D([64, 128]))                        9
    r1_inputs, r1_outputs = siso_repeat(fn, h_n)             10
    r2_inputs, r2_outputs = siso_repeat(fn, h_ndep)          11
    cc_inputs, cc_outputs = concat(2)                        12
                                                             13
    o_inputs["in"].connect(c_outputs["out"])                 14
    r1_inputs["in"].connect(o_outputs["out"])                15
    r2_inputs["in"].connect(o_outputs["out"])                16
    cc_inputs["in0"].connect(r1_outputs["out"])              17
    cc_inputs["in1"].connect(r2_outputs["out"])              18
    return c_inputs, cc_outputs                              19
```

Figure 4: Simple search space showcasing all language components. See also Figure 5.

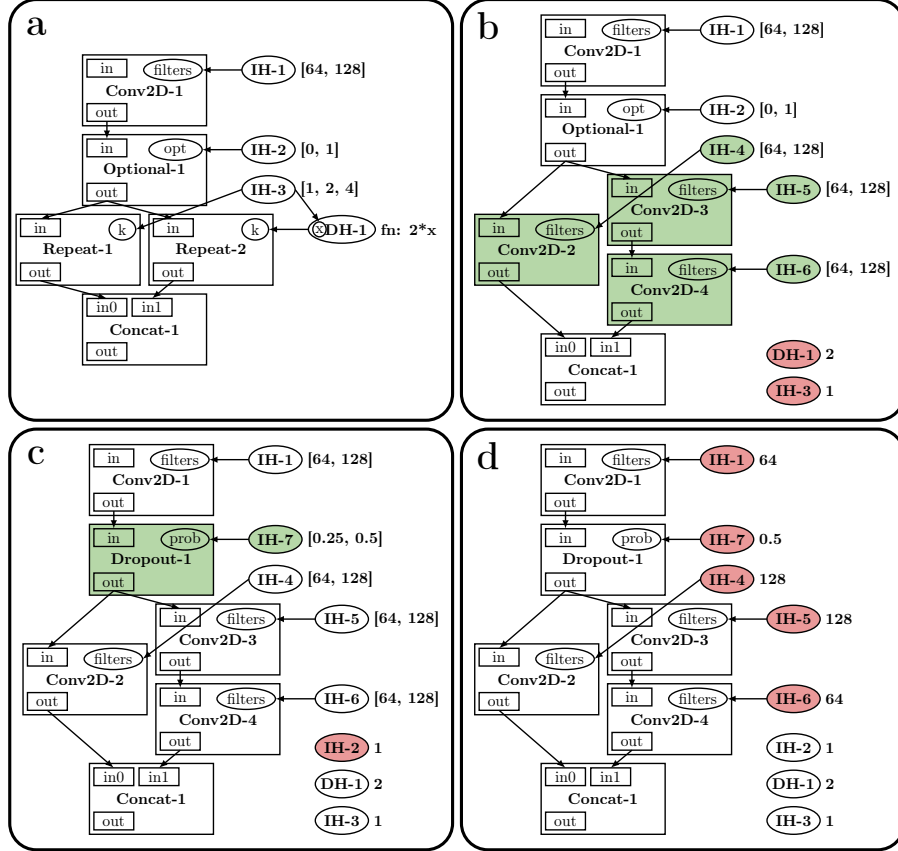

Figure 5: Search space transitions for the search space in Figure 4 (frame a) leading to a single architecture (frame d). Modules and hyperparameters created since the previous frame are highlighted in green. Hyperparameters assigned values since the previous frame are highlighted in red.

rameters types are suffixed with a number to generate unique identifiers. Modules are represented by rectangles that contain inputs, outputs, and properties. Hyperparameters are represented by ellipses (outside of modules) and are associated to module properties (e.g., in frame a, IH-1 is associated to `filters` of `Conv2D-1`). To the right of an independent hyperparameter we show, before assignment, its set of possible values and, after assignment, its value (e.g., IH-1 in frame a and in frame d, respectively). Similarly, for a dependent hyperparameter we show, before assignment, the function that computes its value and, after assignment, its value (e.g., DH-1 in frame a and in frame b, respectively). Frame a shows the initial search space encoded in Figure 4. From frame a to frame b, IH-3 is assigned a value, triggering the value assignment for DH-1 and the substitutions for `Repeat-1` and `Repeat-2`. From frame b to frame c, IH-2 is assigned value 1, creating `Dropout-1` and IH-7 (its dropout rate hyperparameter). Finally, from frame c to frame d, the five remaining independent hyperparameters are assigned values. The search space in frame d has a single architecture that can be mapped to an implementation in a deep learning framework.

# 6 Semantics and mechanics of the search space specification language

In this section, we formally describe the semantics and mechanics of our language and show how they can be used to implement search algorithms for arbitrary search spaces.

## 6.1 Semantics

**Search space components** A search space $G$ has hyperparameters $H(G)$ and modules $M(G)$. We distinguish between independent and dependent hyperparameters as $H_i(G)$ and $H_d(G)$, where

$H(G) = H_i(G) \cup H_d(G)$ and $H_d(G) \cap H_i(G) = \emptyset$, and basic modules and substitution modules as $M_b(G)$ and $M_s(G)$, where $M(G) = M_b(G) \cup M_s(G)$ and $M_b(G) \cap M_s(G) = \emptyset$.

**Hyperparameters** We distinguish between hyperparameters that have been assigned a value and those that have not as $H_a(G)$ and $H_u(G)$. We have $H(G) = H_u(G) \cup H_a(G)$ and $H_u(G) \cap H_a(G) = \emptyset$. We denote the value assigned to an hyperparameter $h \in H_a(G)$ as $v_{(G),(h)} \in \mathcal{X}_{(h)}$, where $h \in H_a(G)$ and $\mathcal{X}_{(h)}$ is the set of possible values for $h$. Independent and dependent hyperparameters are assigned values differently. For $h \in H_i(G)$, its value is assigned directly from $\mathcal{X}_{(h)}$. For $h \in H_d(G)$, its value is computed by evaluating a function $f_{(h)}$ for the values of $H(h)$, where $H(h)$ is the set of hyperparameters that $h$ depends on. For example, in frame a of Figure 5, for $h = \texttt{DH-1}$, $H(h) = \{\texttt{IH-3}\}$. In frame b, $H_a(G) = \{\texttt{IH-3}, \texttt{DH-1}\}$ and $H_u(G) = \{\texttt{IH-1}, \texttt{IH-4}, \texttt{IH-5}, \texttt{IH-6}, \texttt{IH-2}\}$.

**Modules** A module $m \in M(G)$ has inputs $I(m)$, outputs $O(m)$, and hyperparameters $H(m) \subseteq H(G)$ along with mappings assigning names local to the module to inputs, outputs, and hyperparameters, respectively, $\sigma_{(m),i} : S_{(m),i} \to I(m)$, $\sigma_{(m),o} : S_{(m),o} \to O(m)$, $\sigma_{(m),h} : S_{(m),h} \to H(m)$, where $S_{(m),i} \subset \Sigma^*$, $S_{(m),o} \subset \Sigma^*$, and $S_{(m),h} \subset \Sigma^*$, where $\Sigma^*$ is the set of all strings of alphabet $\Sigma$. $S_{(m),i}$, $S_{(m),o}$, and $S_{(m),h}$ are, respectively, the local names for the inputs, outputs, and hyperparameters of $m$. Both $\sigma_{(m),i}$ and $\sigma_{(m),o}$ are bijective, and therefore, the inverses $\sigma_{(m),i}^{-1} : I(m) \to S_{m,i}$ and $\sigma_{(m),o}^{-1} : O(m) \to S_{(m),o}$ exist and assign an input and output to its local name. Each input and output belongs to a single module. $\sigma_{(m),h}$ might not be injective, i.e., $|S_{(m),h}| \geq |H(m)|$. A name $s \in S_{(m),h}$ captures the local semantics of $\sigma_{(m),h}(s)$ in $m \in M(G)$ (e.g., for a convolutional basic module, the number of filters or the kernel size). Given an input $i \in I(M(G))$, $m(i)$ recovers the module that $i$ belongs to (analogously for outputs). For $m \neq m'$, we have $I(m) \cap I(m') = \emptyset$ and $O(m) \cap O(m') = \emptyset$, but there might exist $m, m' \in M(G)$ for which $H(m) \cap H(m') \neq \emptyset$, i.e., two different modules might share hyperparameters but inputs and outputs belong to a single module. We use shorthands $I(G)$ for $I(M(G))$ and $O(G)$ for $O(M(G))$. For example, in frame a of Figure 5, for $m = \texttt{Conv2D-1}$ we have: $I(m) = \{\texttt{Conv2D-1.in}\}$, $O(m) = \{\texttt{Conv2D-1.out}\}$, and $H(m) = \{\texttt{IH-1}\}$; $S_{(m),i} = \{\texttt{in}\}$ and $\sigma_{(m),i}(\texttt{in}) = \texttt{Conv2D-1.in}$ ($\sigma_{(m),o}$ and $\sigma_{(m),h}$ are similar); $m(\texttt{Conv2D-1.in}) = \texttt{Conv2D-1}$. Output and inputs are identified by the global name of their module and their local name within their module joined by a dot, e.g.. $\texttt{Conv2D-1.in}$

**Connections between modules** Connections between modules in $G$ are represented through the set of directed edges $E(G) \subseteq O(G) \times I(G)$ between outputs and inputs of modules in $M(G)$. We denote the subset of edges involving inputs of a module $m \in M(G)$ as $E_i(m)$, i.e., $E_i(m) = \{(o,i) \in E(G) \mid i \in I(m)\}$. Similarly, for outputs, $E_o(m) = \{(o,i) \in E(G) \mid o \in O(m)\}$. We denote the set of edges involving inputs or outputs of $m$ as $E(m) = E_i(m) \cup E_o(m)$. In frame $a$ of Figure 5, For example, in frame a of Figure 5, $E_i(\texttt{Optional-1}) = \{(\texttt{Conv2D-1.out}, \texttt{Optional-1.in})\}$ and $E_o(\texttt{Optional-1}) = \{(\texttt{Optional-1.out}, \texttt{Repeat-1.in}), (\texttt{Optional-1.out}, \texttt{Repeat-2.in})\}$.

**Search spaces** We denote the set of all possible search spaces as $\mathcal{G}$. For a search space $G \in \mathcal{G}$, we define $\mathcal{R}(G) = \{G' \in \mathcal{G} \mid G_1, \ldots, G_m \in \mathcal{G}^m, G_{k+1} = \texttt{Transition}(G_k, h, v), h \in H_i(G_k) \cap H_u(G_k), v \in \mathcal{X}_{(h)}, \forall k \in [m], G_1 = G, G_m = G'\}$, i.e., the set of reachable search spaces through a sequence of value assignments to independent hyperparameters (see Algorithm 1 for the description of $\texttt{Transition}$). We denote the set of terminal search spaces as $\mathcal{T} \subset \mathcal{G}$, i.e. $\mathcal{T} = \{G \in \mathcal{G} \mid H_i(G) \cap H_u(G) = \emptyset\}$. We denote the set of terminal search spaces that are reachable from $G \in \mathcal{G}$ as $\mathcal{T}(G) = \mathcal{R}(G) \cap \mathcal{T}$. In Figure 5, if we let $G$ and $G'$ be the search spaces in frame a and d, respectively, we have $G' \in \mathcal{T}(G)$.

## 6.2 Mechanics

**Search space transitions** A search space $G \in \mathcal{G}$ encodes a set of architectures (i.e., those in $\mathcal{T}(G)$). Different architectures are obtained through different sequences of value assignments leading to search spaces in $\mathcal{T}(G)$. Graph transitions result from value assignments to independent hyperparameters. Algorithm 1 shows how the search space $G' = \texttt{Transition}(G, h, v)$ is computed, where $h \in H_i(G) \cap H_u(G)$ and $v \in \mathcal{X}_{(h)}$. Each transition leads to progressively smaller search spaces (i.e., for all $G \in \mathcal{G}, G' = \texttt{Transition}(G, h, v)$ for $h \in H_i(G) \cap H_u(G)$ and $v \in \mathcal{X}_{(h)}$, then $\mathcal{R}(G') \subseteq \mathcal{R}(G)$). A search space $G' \in \mathcal{T}(G)$ is reached once there are no independent

**Algorithm 1: Transition**

**Input:** $G, h \in H_i(G) \cap H_u(G), v \in \mathcal{X}_{(h)}$

1   $v_{(G),(h)} \leftarrow v$
2   **do**
3      $\tilde{H}_d(G) = \{h \in H_d(G) \cap H_u(G) \mid H_u(h) = \emptyset\}$
4      **for** $h \in \tilde{H}_d(G)$ **do**
5         $n \leftarrow |S_{(h)}|$
6         Let $S_{(h)} = \{s_1, \ldots, s_n\}$ with $s_1 < \ldots < s_n$
7         $v_{(G),(h)} \leftarrow f_{(h)}(v_{G, \sigma_{(h)}(s_1)}, \ldots, v_{G, \sigma_{(h)}(s_n)})$
8      $\tilde{M}_s(G) = \{m \in M_s(G) \mid H_u(m) = \emptyset\}$
9      **for** $m \in \tilde{M}_s(G)$ **do**
10        $n \leftarrow |S_{(m),h}|$
11        Let $S_{(m),h} = \{s_1, \ldots, s_n\}$ with $s_1 < \ldots < s_n$
12        $(G_m, \sigma_i, \sigma_o) = f_{(m)}(v_{G, \sigma_{(m),h}(s_1)}, \ldots, v_{G, \sigma_{(m),h}(s_n)})$
13        $E_i = \{(o, i') \mid (o, i) \in E_i(m), i' = \sigma_i(\sigma_{(m),i}^{-1}(i))\}$
14        $E_o = \{(o', i) \mid (o, i) \in E_o(m), o' = \sigma_o(\sigma_{(m),o}^{-1}(o))\}$
15        $E(G) \leftarrow (E(G) \setminus E(m)) \cup (E_i \cup E_o)$
16        $M(G) \leftarrow (M(G) \setminus \{m\}) \cup M(G_m)$
17        $H(G) \leftarrow H(G) \cup H(G_m)$
18   **while** $\tilde{H}_d(G) \neq \emptyset$ or $\tilde{M}_s(G) \neq \emptyset$;
19   **return** $G$

---

**Algorithm 2: OrderedHyperps**

**Input:** $G, \sigma_o : S_o \to O_u(G)$

1   $M_q \leftarrow \texttt{OrderedModules}(G, \sigma_o)$
2   $H_q \leftarrow [\,]$
3   **for** $m \in M_q$ **do**
4      $n = |S_{(m),h}|$
5      Let $S_{(m),h} = \{s_1, \ldots, s_n\}$ with $s_1 < \ldots < s_n$.
6      **for** $j \in [n]$ **do**
7        $h \leftarrow \sigma_{(m),h}(s_j)$
8        **if** $h \notin H_q$ **then**
9          $H_q \leftarrow H_q + [h]$
10   **for** $h \in H_q$ **do**
11      **if** $h \in H_d(G)$ **then**
12        $n \leftarrow |S_{(h)}|$
13        Let $S_{(h)} = \{s_1, \ldots, s_n\}$ with $s_1 < \ldots < s_n$
14        **for** $j \in [n]$ **do**
15          $h' \leftarrow \sigma_{(h)}(s_j)$
16          **if** $h' \notin H_q$ **then**
17            $H_q \leftarrow H_q + [h']$
18   **return** $H_q$

Figure 6: *Left:* Transition assigns a value to an independent hyperparameter and resolves assignments to dependent hyperparameters (line 3 to 7) and substitutions (line 8 to 17) until none are left (line 18). *Right:* OrderedHyperps returns $H(G)$ sorted according to a unique order. Adds the hyperparameters that are immediately reachable from modules (line 1 to 9), and then traverses the dependencies of the dependent hyperparameters to find additional hyperparameters (line 10 to 17).

hyperparameters left to assign values to, i.e., $H_i(G) \cap H_u(G) = \emptyset$. For $G' \in \mathcal{T}(G)$, $M_s(G') = \emptyset$, i.e., there are only basic modules left. For search spaces $G \in \mathcal{G}$ for which $M_s(G) = \emptyset$, we have $M(G') = M(G)$ (i.e., $M_b(G') = M_b(G)$) and $H(G') = H(G)$ for all $G' \in \mathcal{R}(G)$, i.e., no new modules and hyperparameters are created as a result of graph transitions. Algorithm 1 can be implemented efficiently by checking whether assigning a value to $h \in H_i(G) \cap H_u(G)$ triggered substitutions of neighboring modules or value assignments to neighboring hyperparameters. For example, for the search space $G$ of frame d of Figure 5, $M_s(G) = \emptyset$. Search spaces $G$, $G'$, and $G''$ for frames a, b, and c, respectively, are related as $G' = \texttt{Transition}(G, \texttt{IH-3}, 1)$ and $G'' = \texttt{Transition}(G', \texttt{IH-2}, 1)$. For the substitution resolved from frame b to frame c, for $m = \texttt{Optional-1}$, we have $\sigma_i(\texttt{in}) = \texttt{Dropout-1.in}$ and $\sigma_o(\texttt{out}) = \texttt{Dropout-1.out}$ (see line 12 in Algorithm 1).

**Traversals over modules and hyperparameters**   Search space traversal is fundamental to provide the interface to search spaces that search algorithms rely on (e.g., see Algorithm 3) and to automatically map terminal search spaces to their runnable computational graphs (see Algorithm 4 in Appendix C). For $G \in \mathcal{G}$, this iterator is implemented by using Algorithm 2 and keeping only the hyperparameters in $H_u(G) \cap H_i(G)$. The role of the search algorithm (e.g., see Algorithm 3) is to recursively assign values to hyperparameters in $H_u(G) \cap H_i(G)$ until a search space $G' \in \mathcal{T}(G)$ is reached. Uniquely ordered traversal of $H(G)$ relies on uniquely ordered traversal of $M(G)$. (We defer discussion of the module traversal to Appendix C, see Algorithm 5.)

**Architecture instantiation**   A search space $G \in \mathcal{T}$ can be mapped to a domain implementation (e.g. computational graph in Tensorflow [24] or PyTorch [25]). Only fully-specified basic modules are left in a terminal search space $G$ (i.e., $H_u(G) = \emptyset$ and $M_s(G) = \emptyset$). The mapping from a terminal search space to its implementation relies on graph traversal of the modules according to the topological ordering of their dependencies (i.e., if $m'$ connects to an output of $m$, then $m'$ should be visited after $m$).

Appendix C details this graph propagation process (see Algorithm 4). For example, it is simple to see how the search space of frame d of Figure 5 can be mapped to an implementation.

### 6.3 Supporting search algorithms

Search algorithms interface with search spaces through ordered iteration over unassigned independent hyperparameters (implemented with the help of Algorithm 2) and value assignments to these hyperparameters (which are resolved with Algorithm 1). Algorithms are run for a fixed number of evaluations $k \in \mathbb{N}$, and return the best architecture found. The iteration functionality in Algorithm 2 is independent of the search space and therefore can be used to expose search spaces to search algorithms. We use this decoupling to mix and match search spaces and search algorithms without implementing each pair from scratch (see Section 7).

---

**Algorithm 3:** Random search.

**Input:** $G, \sigma_o : S_o \rightarrow O_u(G), k$

1  $r_{\text{best}} \leftarrow -\infty$
2  **for** $j \in [k]$ **do**
3      $G' \leftarrow G$
4      **while** $G' \notin \mathcal{T}$ **do**
5          $H_q \leftarrow \texttt{OrderedHyperps}(G', \sigma_o)$
6          **for** $h \in H_q$ **do**
7              **if** $h \in H_u(G') \cap H_i(G')$ **then**
8                  $v \sim \texttt{Uniform}(\mathcal{X}_{(h)})$
9                  $G' \leftarrow \texttt{Transition}(G', h, v)$
10     $r \leftarrow \texttt{Evaluate}(G')$
11     **if** $r > r_{best}$ **then**
12         $r_{\text{best}} \leftarrow r$
13         $G_{\text{best}} \leftarrow G'$
14 **return** $G_{best}$

---

Figure 7: Assigns a value uniformly at random (line 8) for each independent hyperparameter (line 7) in the search space until a terminal search space is reached (line 4).

## 7 Experiments

We showcase the modularity and programmability of our language by running experiments that rely on decoupled of search spaces and search algorithms. The interface to search spaces provided by the language makes it possible to reuse implementations of search spaces and search algorithms.

### 7.1 Search space experiments

We vary the search space and fix the search algorithm and the evaluation method. We refer to the search spaces we consider as Nasbench [27], Nasnet [28], Flat [15], and Genetic [26]. For the search phase, we randomly sample 128 architectures from each search space and train them for 25 epochs with Adam with a learning rate of 0.001. The test results for the fully trained architecture with the best validation accuracy are reported in Table 1. These experiments provide a simple characterization of the search spaces in terms of the number of parameters,

Table 1: Test results for search space experiments.

| Search Space | Test Accuracy |
| --- | --- |
| Genetic [26] | 90.07 |
| Flat [15] | 93.58 |
| Nasbench [27] | 94.59 |
| Nasnet [28] | 93.77 |

training times, and validation performances at 25 epochs of the architectures in each search space (see Figure 8). Our language makes these characterizations easy due to better modularity (the implementations of the search space and search algorithm are decoupled) and programmability (new search spaces can be encoded and new search algorithms can be developed).

### 7.2 Search algorithm experiments

We evaluate search algorithms by running them on the same search space. We use the Genetic search space [26] for these experiments as Figure 8 shows its architectures train quickly and have substantially different validation accuracies. We examined the performance of four search algorithms: random, regularized evolution, sequential model based optimization (SMBO), and Monte Carlo Tree Search (MCTS). Random search uniformly samples values for independent hyperparameters (see Algorithm 3). Regularized evolution [14] is an evolutionary algorithm

Table 2: Test results for search algorithm experiments.

| Search algorithm | Test Accuracy |
| --- | --- |
| Random | $91.61 \pm 0.67$ |
| MCTS [29] | $91.45 \pm 0.11$ |
| SMBO [16] | $91.93 \pm 1.03$ |
| Evolution [14] | $91.32 \pm 0.50$ |

that mutates the best performing member of the population and discards the oldest. We use population

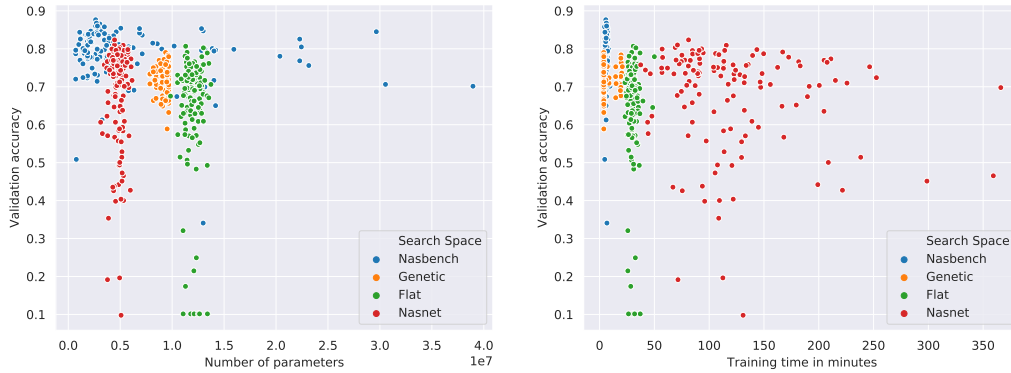

Figure 8: Results for the architectures sampled in the search space experiments. *Left*: Relation between number of parameters and validation accuracy at 25 epochs. *Right*: Relation between time to complete 25 epochs of training and validation accuracy.

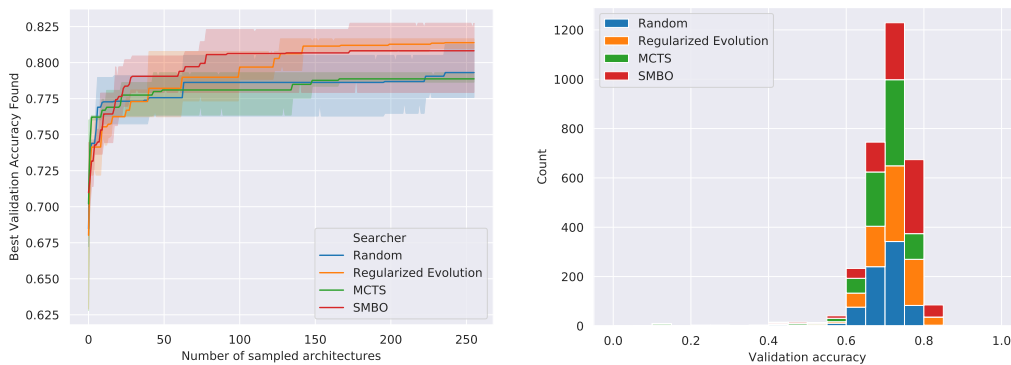

Figure 9: Results for search algorithm experiments. *Left*: Relation between the performance of the best architecture found and the number of architectures sampled. *Right*: Histogram of validation accuracies for the architectures encountered by each search algorithm.

size 100 and sample size 25. For SMBO [16], we use a linear surrogate function to predict the validation accuracy of an architecture from its features (hashed modules sequences and hyperparameter values). For each architecture requested from this search algorithm, with probability $0.1$ a randomly specified architecture is returned; otherwise it evaluates 512 random architectures with the surrogate model and returns the one with the best predicted validation accuracy. MCTS [29, 16] uses the Upper Confidence Bound for Trees (UCT) algorithm with the exploration term of $0.33$. Each run of the search algorithm samples 256 architectures that are trained for 25 epochs with Adam with a learning rate of $0.001$. We ran three trials for each search algorithm. See Figure 9 and Table 2 for the results. By comparing Table 1 and Table 2, we see that the choice of search space had a much larger impact on the test accuracies observed than the choice of search algorithm. See Appendix F for more details.

# 8   Conclusions

We design a language to encode search spaces over architectures to improve the programmability and modularity of architecture search research and practice. Our language allows us to decouple the implementations of search spaces and search algorithms. This decoupling enables to mix-and-match search spaces and search algorithms without having to write each pair from scratch. We reimplement search spaces and search algorithms from the literature and compare them under the same conditions. We hope that decomposing architecture search experiments through the lens of our language will lead to more reusable and comparable architecture search research.

# 9 Acknowledgements

We thank the anonymous reviewers for helpful comments and suggestions. We thank Graham Neubig, Barnabas Poczos, Ruslan Salakhutdinov, Eric Xing, Xue Liu, Carolyn Rose, Zhiting Hu, Willie Neiswanger, Christoph Dann, Kirielle Singajarah, and Zejie Ai for helpful discussions. We thank Google for generous TPU and GCP grants. This work was funded in part by NSF grant IIS 1822831.

## Footnotes

*Part of this work was done while the first author was a research scientist at Petuum.

[2]Visit `https://github.com/negrinho/deep_architect` for code and documentation.

[3]*cf.* the effect of highly programmable deep learning frameworks on deep learning research and practice.

[4]Substitution modules are inspired by delayed evaluation in programming languages.

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
