[Supplementary Material]

# A  Additional details about language components

**Independent hyperparameters**  An hyperparameter can be shared by instantiating it and using it in multiple modules. For example, in Figure 10, `conv_fn` has access to `h_filters` and `h_stride` through a closure and uses them in boths calls. There are 27 architectures in this search space (corresponding to the possible choices for the number of filters, stride, and kernel size). The output of the first convolution is connected to the input of the second through the call to `connect` (line 7).

```
h_filters = D([32, 64, 128])                            1
h_stride = D([1])                                       2
conv_fn = lambda h_kernel_size: conv2d(                 3
    h_filters, h_stride, h_kernel_size)                 4
(c1_inputs, c1_outputs) = conv_fn(D([1, 3, 5]))         5
(c2_inputs, c2_outputs) = conv_fn(D([1, 3, 5]))         6
c1_outputs["out"].connect(c2_inputs["in"])              7
```

Figure 10: Search space with two convolutions in series. The number of filters is the same for both, while the kernel sizes are chosen separately.

**Dependent hyperparameters**  Chains (or general directed acyclic graphs) involving dependent and independent hyperparameters are valid. The search space in Figure 11 has three convolutional modules in series. Each convolutional module shares the hyperparameter for the stride, does not share the hyperparameter for the kernel size, and relates the hyperparameters for the number of filters via a chain of dependent hyperparameters. Each dependent hyperparameter depends on the previous hyperparameter and on the multiplier hyperparameter. This search space has 243 distinct architectures

```
h_filters_lst = [D([32, 64, 128])]                      1
h_factor = D([1, 2, 4])                                 2
h_stride = D([1])                                       3
io_lst = []                                             4
for i in range(3):                                      5
    h = h_filters_lst[i]                                6
    (inputs, outputs) = conv2d(h, h_stride,             7
                               D([1, 3, 5]))            8
    io_lst.append((inputs, outputs))                    9
    if i > 0:                                           10
        io_lst[i - 1][1]["out"].connect(                11
            io_lst[i][0]["in"])                          12
    if i < 2:                                           13
        h_next = DependentHyperparameter(               14
            lambda x, y: x * y,                          15
            {"x": h, "y": h_factor})                     16
        h_filters_lst.append(h_next)                    17
```

Figure 11: Search space with three convolutions in series. The number of filters of an inner convolution is a multiple of the number of filters of the previous convolution. The multiple is chosen through an hyperparameter (`h_factor`).

Encoding this search space in our language might not seem advantageous when compared to encoding it in an hyperparameter optimization tool. Similarly to ours, the latter requires defining hyperparameters for the multiplier, the initial number of filters, and the three kernel sizes (chosen separately). Unfortunately, the encoding by itself tells us nothing about the mapping from hyperparameter values to implementations— the expert must write separate code for this mapping and change it when the search space changes. By contrast, in our language the expert only needs to write the encoding for the search space—the mapping to implementations is induced automatically from the encoding.

```
def dense(h_units):                                     1
    def compile_fn(di, dh):                             2
        m =  tf.layers.Dense(dh['units'])               3
        def forward_fn(di):                             4
            return {"out": m(di["in"])}                 5
        return forward_fn                               6
    name_to_hyperp = {'units': h_units}                 7
    return siso_tensorflow_module(                      8
        'Affine', compile_fn, name_to_hyperp, scope)    9
```

```
def conv2d(h_num_filters, h_filter_width, h_stride):    1
    def compile_fn(di, dh):                             2
        conv_op = tf.layers.Conv2D(                     3
            dh['num_filters'],                           4
            (dh['filter_width'],) * 2,                   5
            (dh['stride'],) * 2,                         6
            padding='SAME')                              7
        def forward_fn(di):                             8
            return {'out': conv_op(di['in'])}           9
        return forward_fn                               10
    return siso_tensorflow_module(                      11
        'Conv2D', compile_fn, {                          12
            'num_filters': h_num_filters,                13
            'filter_width': h_filter_width,              14
            'stride': h_stride                           15
        })                                              16
```

Figure 12: Examples of basic modules in our implementation resulting from wrapping Tensorflow operations. *Left:* Affine basic module with an hyperparameter for the number of units. *Right:* Convolutional basic module with hyperparameters for the number of filters, filter size, and stride.

**Basic Modules**  Deep learning layers can be easily wrapped as basic modules. For example, a dense layer can be wrapped as a single-input single-output module with one hyperparameter for the number of units (see left of Figure 12). A convolutional layer is another example of a

single-input single-output module (see right of Figure 12). The implementation of `conv2d` relies on `siso_tensorflow_module` for wrapping Tensorflow-specific aspects (see Appendix E.1 for a discussion on how to support different domains). `conv2d` depends on hyperparameters for `num_filters`, `filter_width`, and `stride`. The key observation is that a basic module generates its implementation (calls to `compile_fn` and then `forward_fn`) only after its hyperparameter values have been assigned and it has values for its inputs. The values of the inputs and the hyperparameters are available in the dictionaries `di` and `dh`, respectively. `conv2d` returns a module as (`inputs`, `outputs`) (these are analogous to $\sigma_i$ and $\sigma_h$ on line of 12 of Algorithm 1). Instantiating the computational graph relies on `compile_fn` and `forward_fn`. `compile_fn` is called a single time, e.g., to instantiate the parameters of the basic module. `forward_fn` can be called multiple times to create the computational graph (in static frameworks such as Tensorflow) or to evaluate the computational graph for specific data (e.g., in dynamic frameworks such as PyTorch). Parameters instantiated in `compile_fn` are available to `forward_fn` through a closure.

```
def mimo_or(fn_lst, h_or, input_names,           1
        output_names, scope=None, name=None):    2
    def substitution_fn(dh):                     3
        return fn_lst[dh["idx"]]()               4
                                                 5
    return substitution_module(                  6
        _get_name(name, "Or"),                   7
        substitution_fn,                         8
        {'idx': h_or},                           9
        input_names, output_names, scope)        10
```

```
def siso_repeat(fn, h_num_repeats,               1
        scope=None, name=None):                  2
    def substitution_fn(dh):                     3
        assert dh["num_reps"] > 0                4
        return siso_sequential([fn()             5
            for _ in range(dh["num_reps"])])     6
                                                 7
    return substitution_module(                  8
        _get_name(name, "SISORepeat"),           9
        substitution_fn,                         10
        {'num_reps': h_num_repeats},             11
        ['in'], ['out'], scope)                  12
```

```
def siso_split_combine(fn, combine_fn,                      1
        h_num_splits, scope=None, name=None):               2
    def substitution_fn(dh):                                3
        inputs_lst, outputs_lst = zip(*[fn()                4
            for _ in range(dh["num_splits"])])              5
        c_inputs, c_outputs = combine_fn(                   6
            dh["num_splits"])                               7
                                                            8
        i_inputs, i_outputs = identity()                    9
        for i in range(dh["num_splits"]):                   10
            i_outputs['out'].connect(                       11
                inputs_lst[i]['in'])                        12
            c_inputs['in' + str(i)].connect(                13
                outputs_lst[i]['out'])                      14
        return i_inputs, c_outputs                          15
                                                            16
    return substitution_module(                             17
        _get_name(name, "SISOSplitCombine"),                18
        substitution_fn,                                    19
        {'num_splits': h_num_splits},                       20
        ['in'], ['out'], scope)                             21
```

Figure 13: Example substitution modules implemented in our framework. *Top left:* `mimo_or` chooses between a list of functions returning search spaces. *Bottom left:* Creates a series connection of the search space returned by `fn` some number of times (determined by `h_num_repeats`). *Right:* Creates a search space with a number (determined by `h_num_splits`) of single-input single-output parallel search spaces created by `fn` that are then combined into the search space created by `combine_fn`.

**Substitution modules** Substitution modules encode local structural transformations of the search space that are resolved once all their hyperparameters have been assigned values (see line 12 in Algorithm 1). Consider the implementation of `mimo_or` (i.e., mimo stands for multi-input, multi-output) in Figure 13 (top left). We make substantial use of higher-order functions and closures in our language implementation. For example, to implement a specific `or` substitution module, we only need to provide a list of functions that return search spaces. Arguments that the functions would need to carry are accessed through the closure or through argument binding[5]. `mimo_or` has an hyperparameter for which subsearch space function to pick (`h_idx`). Once `h_idx` is assigned a value, `substitution_fn` is called, returning a search space as (`inputs`, `outputs`) where `inputs` and `outputs` are $\sigma_i$ and $\sigma_o$ mentioned on line 12 of Algorithm 1. Using mappings of inputs and outputs is convenient because it allow us to treat modules and search spaces the same (e.g., when connecting search spaces). The other substitution modules in Figure 13 use `substitution_fn` similarly.

Figure 14 shows the signature of the wrapper function to easily create substitution modules. All information about what subsearch space should be generated upon substitution is delegated to `substitution_fn`. Compare this to signature of `keras_module` for Keras basic modules in Figure 21.

```
def substitution_module(name, name_to_hyperp,        1
        substitution_fn, input_names, output_names):  2
```

Figure 14: Signature of the helper used to create substitution modules.

**Auxiliary functions**   Figure 15 shows how we often design search spaces. We have a high-level inductive bias (e.g., what operations are likely to be useful) for a good architecture for a task, but we might be unsure about low-level details (e.g., the exact sequence of operations of the architecture). Auxiliary functions allows us to encapsulate aspects of search space creation and can be reused for creating different search spaces, e.g., through different calls to these functions.

$$i_t = \sigma(W_{ii}x_t + b_{ii} + W_{hi}h_{t-1} + b_{hi})$$
$$f_t = \sigma(W_{if}x_t + b_{if} + W_{hf}h_{t-1} + b_{hf})$$
$$g_t = \tanh(W_{ig}x_t + b_{ig} + W_{hg}h_{t-1} + b_{hg})$$
$$o_t = \sigma(W_{io}x_t + b_{io} + W_{ho}h_{t-1} + b_{ho})$$
$$c_t = f_t c_{t-1} + i_t g_t$$
$$h_t = o_t \tanh(c_t)$$

$$i_t = q_i(x_t, h_{t-1})$$
$$f_t = q_f(x_t, h_{t-1})$$
$$g_t = q_g(x_t, h_{t-1})$$
$$o_t = q_o(x_t, h_{t-1})$$
$$c_t = q_c(f_t, c_{t-1}, i_t, g_t)$$
$$h_t = q_h(o_t, c_t)$$

```
def lstm_cell(input_fn, forget_fn, gate_fn,          1
              output_fn, cell_fn, hidden_fn):        2
                                                     3
    x_inputs, x_outputs = identity()                 4
    hprev_inputs, hprev_outputs = identity()         5
    cprev_inputs, cprev_outputs = identity()         6
                                                     7
    i_inputs, i_outputs = input_fn()                 8
    f_inputs, f_outputs = forget_fn()                9
    g_inputs, g_outputs = gate_fn()                  10
    o_inputs, o_outputs = output_fn()                11
    c_inputs, c_outputs = cell_fn()                  12
    h_inputs, h_outputs = hidden_fn()                13
                                                     14
    i_inputs["in0"].connect(x_outputs["out"])        15
    i_inputs["in1"].connect(hprev_outputs["out"])    16
    f_inputs["in0"].connect(x_outputs["out"])        17
    f_inputs["in1"].connect(hprev_outputs["out"])    18
    g_inputs["in0"].connect(x_outputs["out"])        19
    g_inputs["in1"].connect(hprev_outputs["out"])    20
    c_inputs["in0"].connect(f_outputs["out"])        21
    c_inputs["in1"].connect(cprev_outputs["out"])    22
    c_inputs["in2"].connect(i_outputs["out"])        23
    c_inputs["in3"].connect(g_outputs["out"])        24
    o_inputs["in0"].connect(x_inputs["in"])          25
    o_inputs["in1"].connect(hprev_inputs["in"])      26
    h_inputs["in0"].connect(o_outputs["out"])        27
    h_inputs["in1"].connect(c_outputs["out"])        28
                                                     29
    return ({"x": x_inputs["in"],                    30
             "hprev": hprev_inputs["in"],            31
             "cprev": cprev_inputs["in"]},           32
            {"c": c_outputs["out"],                  33
             "h": h_outputs["out"]})                 34
```

Figure 15: *Left:* LSTM equations showing how the expert might abstract the LSTM structure into a general functional dependency. *Right:* Auxiliary function for a LSTM cell that takes functions that return the search spaces for input, output, and forget gates, and the cell update, hidden state output, and context mechanisms and arranges them together to create the larger LSTM-like search space.

## B   Search space example

Figure 16 shows the recurrent cell search space introduced in [23] encoded in our language implementation. This search space is composed of a sequence of nodes. For each node, we choose its type and from which node output will it get its input. The cell output is the average of the outputs of all nodes after the first one. The encoding of this search space exemplifies the expressiveness of substitution modules. The cell connection structure is created through a substitution module that has hyperparameters representing where each node will get its input from. The substitution function that creates this cell takes functions that return inputs and outputs of the subsearch spaces for the input and intermediate nodes. Each subsearch space determines the operation performed by the node. While more complex than the other examples that we have presented, the same language constructs allow us to approach the encoding of this search space. Functions `cell`, `input_node`, `intermediate_node`, and `search_space` define search spaces that are fully encapsulated and that therefore, can be reused for creating new search spaces.

## C   Additional details about language mechanics

**Ordered module traversal**   Algorithm 5 generates a unique ordering over modules $M(G)$ by starting at the modules that have outputs in $O_u(G)$ (which are named by $\sigma_o$) and traversing backwards, moving from a module to its neighboring modules (i.e., the modules that connect an output to an input of this module). A unique ordering is generated by relying on the lexicographic ordering of the local names (see lines 3 and 10 in Algorithm 5).

```
def cell(num_nodes,                          1
         h_units,                            2
         input_node_fn,                      3
         intermediate_node_fn,               4
         combine_fn):                        5
                                             6
    def substitution_fn(dh):                 7
        input_node = input_node_fn(h_units)  8
        inter_nodes = [                      9
            intermediate_node_fn(h_units)    10
            for _ in range(1, num_nodes)     11
        ]                                    12
        nodes = [input_node] + inter_nodes   13
                                             14
        for i in range(1, num_nodes):        15
            nodes[i][0]["in"].connect(       16
                nodes[dh[str(i)]][1]["out"]) 17
                                             18
        used_ids = set(dh.values())          19
        unused_ids = set(range(num_nodes)    20
            ).difference(used_ids)           21
        c_inputs, c_outputs = combine_fn(    22
            len(unused_ids))                 23
        for j, i in enumerate(sorted(unused_ids)): 24
            c_inputs ["in%d"%j].connect(     25
                nodes[i][1]["out"])          26
                                             27
        return (input_node[0],               28
            {"ht+1": c_outputs["out"]})      29
                                             30
    name_to_hyperp = {str(i): D(range(i))    31
        for i in range(1, num_nodes)}        32
                                             33
    return substitution_module("Cell",       34
        substitution_fn, name_to_hyperp,     35
        ["x", "ht"], ["ht+1"])               36
```

```
def input_node_fn(h_units):                              1
    h_inputs, h_outputs = affine(h_units)                2
    x_inputs, x_outputs = affine(h_units)                3
    a_inputs, a_outputs = add(2)                         4
    n_inputs, n_outputs = nonlinearity(D(["relu",        5
        "tanh","sigmoid", "identity"]))                  6
                                                         7
    a_inputs["in0"].connect(x_outputs["out"])            8
    a_inputs["in1"].connect(h_outputs["out"])            9
    n_inputs["in"].connect(a_outputs["out"])             10
                                                         11
    return {                                             12
        "x": x_inputs["in"],                             13
        "ht": h_inputs["in"]}, n_outputs                 14
                                                         15
                                                         16
def intermediate_node_fn(h_units):                       17
    a_inputs, a_outputs = affine(h_units)                18
    n_inputs, n_outputs = nonlinearity(D(["relu",        19
        "tanh", "sigmoid", "identity"]))                 20
    a_outputs["out"].connect(n_inputs["in"])             21
    return a_inputs, n_outputs                           22
```

```
def search_space():                                      1
    h_units = D([32, 64, 128, 256])                      2
    return cell(8, h_units,                              3
        input_node_fn, intermediate_node_fn, avg)        4
```

Figure 16: Recurrent search space from ENAS [23] encoded using our language implementation. A substitution module is used to delay the creation of the cell topology. The code uses higher order functions to create the cell search space from the subsearch spaces of its nodes (i.e., `input_node_fn` and `intermediate_node_fn`).

**Architecture instantiation**    Mapping an architecture $G \in \mathcal{T}$ relies on traversing $M(G)$ in topological order. Intuitively, to do the local computation of a module $m \in M(G)$ for $G \in \mathcal{T}$, the modules that $m$ depends on (i.e., which feed an output into an input of $m$) must have done their local computations to produce their outputs (which will now be available as inputs to $m$). Graph propagation (Algorithm 4) starts with values for the unconnected inputs $I_u(G)$ and applies local module computation according to the topological ordering of the modules until the values for the unconnected outputs $O_u(G)$ are generated. $g_{(m)}$ maps input and hyperparameter values to the local computation of $m$. The arguments of $g_{(m)}$ and its results are sorted according to their local names (see lines 2 to 8).

# D    Discussion about language expressivity

## D.1    Infinite search spaces

We can rely on the laziness of substitution modules to encode infinite search spaces. Figure 18 shows an example of such a search space. If the hyperparameter associated to the substitution module is assigned the value one, a new substitution module and hyperparameter are created. If the hyperparameter associated to the substitution module is assigned the value zero, recursion stops. The search space is infinite because the recursion can continue indefinitely. This search space can be used to create other search spaces compositionally. The same principles are valid for more complex search spaces involving recursion.

```
def maybe_one_more(fn):                              1
    return siso_or([                                 2
        fn, lambda: siso_sequential(                 3
            [fn(), maybe_one_more(fn)])],            4
        D([0, 1]))                                   5
```

Figure 18: Self-similar search space either returns a search space or a search space and an optional additional search space. `fn` returns the search space to use in this construction.

**Algorithm 4:** Forward

**Input:** $G \in \mathcal{T}, x_{(i)}$ for $i \in I_u(G)$ and
$\qquad x_{(i)} \in \mathcal{X}_{(i)}$

1 **for** $m \in OrderedTopologically(M(G))$ **do**
2 $\quad S_{(m),h} = \{s_{h,1}, \ldots, s_{h,n_h}\}$ for
$\qquad s_{h,1} < \ldots < s_{h,n_h}$
3 $\quad S_{(m),i} = \{s_{i,1}, \ldots, s_{i,n_i}\}$ for
$\qquad s_{i,1} < \ldots < s_{i,n_i}$
4 $\quad S_{(m),o} = \{s_{o,1}, \ldots, s_{o,n_o}\}$ for
$\qquad s_{o,1} < \ldots < s_{o,n_o}$
5 $\quad x_j \leftarrow x_{(\sigma_{(m),i}(s_{i,j}))}$, for $j \in [n_i]$
6 $\quad v_j \leftarrow v_{(\sigma_{(m),h}(s_{h,j}))}$, for $j \in [n_h]$
7 $\quad (y_1, \ldots, y_{n_o}) \leftarrow$
$\qquad g_{(m)}(x_1, \ldots, x_{n_i}, v_1, \ldots, v_{n_h})$
8 $\quad y_{\sigma_{(m),o}(s_{o,j})} \leftarrow y_j$ for $j \in [n_o]$
9 $\quad$ **for** $(o, i) \in E_o(m)$ **do**
10 $\qquad x_{(i)} \leftarrow y_{(o)}$

11 **return** $y_{(o)}$ for $o \in O_u(G)$

---

**Algorithm 5:** OrderedModules

**Input:** $G, \sigma_o : S_o \rightarrow O_u(G)$

1 $M_q \leftarrow []$
2 $n \leftarrow |S_o|$
3 Let $S_o = \{s_1, \ldots, s_n\}$ with $s_1 < \ldots < s_n$.
4 **for** $k \in [n]$ **do**
5 $\quad m \leftarrow m(\sigma_o(s_k))$
6 $\quad$ **if** $m \notin M_q$ **then**
7 $\qquad M_q \leftarrow M_q + [m]$

8 **for** $m \in M_q$ **do**
9 $\quad n \leftarrow |S_{(m),i}|$
10 $\quad$ Let $S_{(m),i} = \{s_1, \ldots, s_n\}$ with
$\qquad s_1 < \ldots < s_n$.
11 $\quad$ **while** $j \in [n]$ **do**
12 $\qquad i \leftarrow \sigma_{(m),i}(s_j)$
13 $\qquad$ **if** $i \notin I_u(G)$ **then**
14 $\qquad\quad$ Take $(o, i) \in E(G)$
15 $\qquad\quad m' \leftarrow m(o)$
16 $\qquad\quad$ **if** $m' \notin M_q$ **then**
17 $\qquad\qquad M_q \leftarrow M_q + [m']$

18 **return** $M_q$

Figure 17: *Left:* Forward maps a terminal search space to its domain implementation. The mapping relies on each basic module doing its local computation (encapsulated by $g_{(m)}$ on line 7). Forward starts with values for the unconnected inputs and traverses the modules in topological order to generate values for the unconnected outputs. *Right:* Iteration of $M(G)$ according to a unique order. The first while (line 4) loop adds the modules of the outputs in $O_u(G)$. The second while (line 8) loop traverses backwards the connections of the modules in $M_q$, adding new modules reached this way to $M_q$. $m(o)$ denotes the module that $o$ belongs to. See also Figure 6

## D.2 Search space transformation and combination

We assume the existence of functions a_fn, b_fn, and c_fn that each take one binary hyperparameter and return a search space. In Figure 19, search_space_1 repeats a choice between a_fn, b_fn, and c_fn one, two, or four times. The hyperparameters for the choice (i.e., those associated to siso_or) modules are assigned values separately for each repetition. The hyperparameters associated to each a_fn, b_fn, or c_fn are also assigned values separately.

Simple rearrangements lead to dramatically different search spaces. For example, we get search_space_2 by swapping the nesting order of siso_repeat and siso_or. This search space chooses between a repetition of one, two, or four a_fn, b_fn, or c_fn. Each binary hyperparameter of the repetitions is chosen separately. search_space_3 shows that it is simple to share an hyperparameter across the repetitions by instantiating it outside the function (line 2), and access it on the function (lines 5, 7, and 9). search_space_1, search_space_2, and search_space_3 are encapsulated and can be used as any other search space. search_space_4 shows that we can easily use search_space_1, search_space_2, and search_space_3 in a new search space (compare to search_space_2).

Highly-conditional search spaces can be created through local composition of modules, reducing cognitive load. In our language, substitution modules, basic modules, dependent hyperparameters, and independent hyperparameters are well-defined constructs to encode complex search spaces. For example, a_fn might be complex, creating many modules and hyperparameters, but its definition encapsulates all this. This is one of the greatest advantages of our language, allowing us to easily create new search spaces from existing search spaces. Furthermore, the mapping from instances in the search space to implementations is automatically generated from the search space encoding.

```
def search_space_1():                                   1
    return siso_repeat(                                 2
        lambda: siso_or([                               3
            lambda: a_fn(D([0, 1])),                    4
            lambda: b_fn(D([0, 1])),                    5
            lambda: c_fn(D([0, 1]))],                   6
        D([0, 1, 2])), D([1, 2, 4]))                    7
```

```
def search_space_3():                                   1
    h = D([0, 1])                                       2
    return siso_or([                                    3
        lambda: siso_repeat(                            4
            lambda: a_fn(h), D([1, 2, 4])),             5
        lambda: siso_repeat(                            6
            lambda: b_fn(h), D([1, 2, 4])),             7
        lambda: siso_repeat(                            8
            lambda: c_fn(h), D([1, 2, 4]))],            9
        D([0, 1, 2]))                                   10
```

```
def search_space_2():                                   1
    return siso_or([                                    2
        lambda: siso_repeat(                            3
            lambda: a_fn(D([0, 1])),                    4
                D([1, 2, 4])),                          5
        lambda: siso_repeat(                            6
            lambda: b_fn(D([0, 1])),                    7
                D([1, 2, 4])),                          8
        lambda: siso_repeat(                            9
            lambda: c_fn(D([0, 1])),                    10
                D([1, 2, 4]))],                         11
        D([0, 1, 2]))                                   12
```

```
def search_space_4():                                   1
    return siso_or([                                    2
        lambda: siso_repeat(                            3
            search_space_1, D([1, 2, 4])),              4
        lambda: siso_repeat(                            5
            search_space_2, D([1, 2, 4])),              6
        lambda: siso_repeat(                            7
            search_space_3, D([1, 2, 4]))],             8
        D([0, 1, 2]))                                   9
```

Figure 19: *Top left:* Repeats the choice between `a_fn`, `b_fn`, and `c_fn` one, two, or four times. This search space shows that expressive search spaces can be created through simple arrangements of substitution modules. *Bottom left:* Simple transformation of `search_space_1`. *Top right:* Similar to `search_space_2`, but with the binary hyperparameter shared across all repetitions. *Bottom right:* Simple search space that is created by composing the previously defined search spaces to create a new substitution module.

# E   Implementation details

This section gives concrete details about our Python language implementation. We refer the reader to `https://github.com/negrinho/deep_architect` for additional code and documentation.

## E.1   Supporting new domains

We only need to extend `Module` class to support basic modules in the new domain. We start with the common implementation of `Module` (see Figure 20) for both basic and substitution modules and then cover its extension to support Keras basic modules (see Figure 21).

**General module class**   The complete implementation of `Module` is shown in Figure 20. `Module` supports the implementations of both basic modules and substitution modules. There are three types of functions in `Module` in Figure 20: those that are used by both basic and substitution modules (`_register_input`, `_register_output`, `_register_hyperparameter`, `_register`, `_get_hyperp_values`, `get_io` and `get_hyperps`); those that are used just by basic modules (`_get_input_values`, `_set_output_values`, `_compile`, `_forward`, and `forward`); those are used just by substitution modules (`_update`). We will mainly discuss its extension for basic modules as substitution modules are domain-independent (e.g., there are no domain-specific components in the substitution modules in Figure 13 and in `cell` in Figure 16).

Supporting basic modules in a domain relies on two functions: `_compile` and `_forward`. These functions help us map an architecture to its implementation in deep learning (slightly different functions might be necessary for other domains). `forward` shows how `_compile` and `_forward` are used during graph instantiation in a terminal search space. See Figure 22 for the iteration over the graph in topological ordering (determined by `determine_module_eval_seq`), and evaluates the forward calls in turn for the modules in the graph leading to its unconnected outputs.

`_register_input`, `_register_output`, `_register_hyperparameter`, and `_register` are used to describe the inputs and outputs of the module (i.e., `_register_input` and `_register_output`), and to associate hyperparameters to its properties (i.e., `_register_hyperparameter`). `_register` aggregates the first three functions into one. `_get_hyperp_values`, `_get_input_values`, and `_set_output_values` are used in `_forward` (see left of Figure 21. These are used in each basic module, once in a terminal search space, to retrieve its hyperparameter values (`_get_hyperp_values`) and its input values (`_get_input_values`) and to write the results of its local computation to its outputs (`_set_output_values`). Finally, `get_io` retrieves the dictio-

```
class Module(Addressable):                                                              1
                                                                                        2
    def __init__(self, scope=None, name=None):                                          3
        scope = scope if scope is not None else Scope.default_scope                      4
        name = scope.get_unused_name('.'.join(                                           5
            ['M', (name if name is not None else self._get_base_name()) + '-']))         6
        Addressable.__init__(self, scope, name)                                          7
                                                                                        8
        self.inputs = OrderedDict()                                                      9
        self.outputs = OrderedDict()                                                     10
        self.hyperps = OrderedDict()                                                     11
        self._is_compiled = False                                                        12
                                                                                        13
    def _register_input(self, name):                                                     14
        assert name not in self.inputs                                                   15
        self.inputs[name] = Input(self, self.scope, name)                                16
                                                                                        17
    def _register_output(self, name):                                                    18
        assert name not in self.outputs                                                  19
        self.outputs[name] = Output(self, self.scope, name)                              20
                                                                                        21
    def _register_hyperparameter(self, name, h):                                         22
        assert isinstance(h, Hyperparameter) and name not in self.hyperps               23
        self.hyperps[name] = h                                                           24
        h._register_module(self)                                                         25
                                                                                        26
    def _register(self, input_names, output_names, name_to_hyperp):                      27
        for name in input_names:                                                         28
            self._register_input(name)                                                   29
        for name in output_names:                                                        30
            self._register_output(name)                                                  31
        for name in sorted(name_to_hyperp):                                              32
            self._register_hyperparameter(name, name_to_hyperp[name])                    33
                                                                                        34
    def _get_input_values(self):                                                         35
        return {name: ix.val for name, ix in iteritems(self.inputs)}                      36
                                                                                        37
    def _get_hyperp_values(self):                                                        38
        return {name: h.get_value() for name, h in iteritems(self.hyperps)}             39
                                                                                        40
    def _set_output_values(self, output_name_to_val):                                    41
        for name, val in iteritems(output_name_to_val):                                  42
            self.outputs[name].val = val                                                 43
                                                                                        44
    def get_io(self):                                                                    45
        return self.inputs, self.outputs                                                 46
                                                                                        47
    def get_hyperps(self):                                                               48
        return self.hyperps                                                              49
                                                                                        50
    def _update(self):                                                                   51
        """Called when an hyperparameter that the module depends on is set."""           52
        raise NotImplementedError                                                        53
                                                                                        54
    def _compile(self):                                                                  55
        raise NotImplementedError                                                        56
                                                                                        57
    def _forward(self):                                                                  58
        raise NotImplementedError                                                        59
                                                                                        60
    def forward(self):                                                                   61
        if not self._is_compiled:                                                        62
            self._compile()                                                              63
            self._is_compiled = True                                                     64
        self._forward()                                                                  65
```

Figure 20: Module class used to implement both basic and substitution modules. `_register_input`, `_register_output`, `_register_hyperparameter`, `_register`, `_get_hyperp_values`, `get_io` and `get_hyperps` are used by both basic and substitution modules. `_get_input_values`, `_set_output_values`, `_compile`, `_forward`, and `forward` are used only by basic modules. `_update` is used only by substitution modules.

naries mapping names to inputs and outputs (these correspond to $\sigma_{(m),i} : S_{(m),i} \rightarrow I(m)$ and $\sigma_{(m),o} : S_{(m),o} \rightarrow O(m)$, respectively, described in Section 6). Most inputs are named `in` if there is a single input and `in0`, `in1`, and so on if there is more than one. Similarly, for outputs, we have `out` for a single output, and `out0`, `out1`, and so if there are multiple outputs. This is often seen when connecting search spaces, e.g., lines 15 to 28 in right of Figure 15. In Figure 15, we redefine $\sigma_i$ and $\sigma_o$ (in line 30 to line 34) to have appropriate names for the LSTM cell, but often, if possible, we just use $\sigma_{(m),i}$ and $\sigma_{(m'),o}$ for $\sigma_i$ and $\sigma_o$ respectively, e.g., in `siso_repeat` and `siso_combine` in Figure 13.

`_update` is used in substitution modules (not shown in Figure 20): for a substitution module, it checks if all its hyperparameters have been assigned values and does the substitution (i.e., calls its substitution function to create a search space that takes the place of the substitution module; e.g., see frames a, b, and c of Figure 5 for a pictorial representation, and Figure 13 for implementations of substitution modules). In the examples of Figure 13, `substitution_fn` returns the search space to replace the substitution module with in the form of a dictionary of inputs and a dictionary of outputs (corresponding to $\sigma_i$ and $\sigma_o$ on line 12 of Algorithm 1). The substitution modules that we considered can be implemented with the helper in Figure 14 (e.g., see the examples in Figure 13).

In the signature of `__init__` for `Module`, `scope` is a namespace used to register a module with a unique name and `name` is the prefix used to generate the unique name. Hyperparameters also have a unique name generated in the same way. Figure 5 shows this in how the modules and hyperparameters are named, e.g., in frame a, `Conv2D-1` results from generating a unique identifier for `name Conv2D` (this is also captured in the use of `_get_name` in the examples in Figure 12 and Figure 13). When `scope` is not mentioned explicitly, a default global scope is used (e.g., `scope` is optional in Figure 20).

**Extending the module class for a domain (e.g., Keras)** Figure 21 (left) shows the extension of `Module` to deal with basic modules in Keras. `KerasModule` is the extension of `Module`. `keras_module` is a convenience function that instantiates a `KerasModule` and returns its dictionary of local names to inputs and outputs. `siso_keras_module` is the same as `keras_module` but uses default names `in` and `out` for a single-input single-output module, which saves the expert the trouble of explicitly naming inputs and outputs for this common case. Finally, `siso_keras_module_from_keras_layer_fn` reduces the effort of creating basic modules from Keras functions (i.e., the function can be passed directly creating `compile_fn` beforehand). These functions are analogous for different deep learning frameworks, e.g., see the example usage of `siso_tensorflow_module` in Figure 12.

The most general helper, `keras_module` works by providing the local names for the inputs (`input_names`) and outputs (`output_names`), the dictionary mapping local names to hyperparameters (`name_to_hyperp`), and the compilation function (`compile_fn`), which corresponds to the `_compile_fn` function of the module. Calling `_compile_fn` returns a function (corresponding to `_forward` for a module, e.g., see Figure 12).

## E.2 Implementing a search algorithm

Figure 23 shows random search in our implementation. `random_specify_hyperparameter` assigns a value uniformly at random to an independent hyperparameter. `random_specify` assigns all unassigned independent hyperparameters in the search space until reaching a terminal search space (each assignment leads to a search space transition; see Figure 5). `RandomSearcher` encapsulates the behavior of the searcher through two main functions: `sample` and `update`. `sample` samples an architecture from the search space, which returns `inputs` and `outputs` for the sampled terminal search space, the sequence of value assignments that led to the sampled terminal search space, and a `searcher_eval_token` that allows the searcher to identify the sampled terminal search space when the evaluation results are passed back to the searcher through a call to `update`. `update` incorporates the evaluation results (e.g., validation accuracy) of a sampled architecture into the state of the searcher, allowing it to use this information in the next call to `sample`. For random search, `update` is a no-op. `__init__` takes the function returning a search space (e.g., `search_space` in Figure 16) from which architectures are to be drawn from and any other arguments that the searcher may need (e.g., exploration term in MCTS). To implement a new searcher, `Searcher` needs to be extended by implementing `sample` and `update` for the desired search algorithm. `unassigned_independent_hyperparameter_iterator` provides ordered iteration over

```python
import deep_architect.core as co

class KerasModule(co.Module):

    def __init__(self,
                 name,
                 compile_fn,
                 name_to_hyperp,
                 input_names,
                 output_names,
                 scope=None):
        co.Module.__init__(self, scope, name)
        self._register(input_names, output_names,
            name_to_hyperp)
        self._compile_fn = compile_fn

    def _compile(self):
        input_name_to_val = self._get_input_values()
        hyperp_name_to_val = self._get_hyperp_values()
        self._fn = self._compile_fn(
            input_name_to_val, hyperp_name_to_val)

    def _forward(self):
        input_name_to_val = self._get_input_values()
        output_name_to_val = self._fn(input_name_to_val)
        self._set_output_values(output_name_to_val)

    def _update(self):
        pass
```

```python
def keras_module(name,
                 compile_fn,
                 name_to_hyperp,
                 input_names,
                 output_names,
                 scope=None):
    return KerasModule(name, compile_fn,
        name_to_hyperp, input_names,
        output_names, scope).get_io()

def siso_keras_module(name, compile_fn,
        name_to_hyperp, scope=None):
    return KerasModule(name, compile_fn,
        name_to_hyperp, ['in'], ['out'],
                        scope).get_io()

def siso_keras_module_from_keras_layer_fn(
        layer_fn, name_to_hyperp,
        scope=None, name=None):

    def compile_fn(di, dh):
        m = layer_fn(**dh)

        def forward_fn(di):
            return {"out": m(di["in"])}

        return forward_fn

    if name is None:
        name = layer_fn.__name__

    return siso_keras_module(name,
        compile_fn, name_to_hyperp, scope)
```

Figure 21: *Left:* Complete extension of the Module class (see Figure 20 for supporting Keras basic modules. *Right:* Convenience functions to reduce the effort of wrapping Keras operations into basic modules for common cases. See Figure 12 for examples of how they are used.

```python
def forward(input_to_val, _module_seq=None):
    if _module_seq is None:
        _module_seq = determine_module_eval_seq(input_to_val.keys())

    for ix, val in iteritems(input_to_val):
        ix.val = val

    for m in _module_seq:
        m.forward()
        for ox in itervalues(m.outputs):
            for ix in ox.get_connected_inputs():
                ix.val = ox.val
```

Figure 22: Generating the implementation of the architecture in a terminal search space $G$ (e.g., the one in frame d of Figure 5). Compare to Algorithm 4: input_to_val corresponds to the $x_{(i)}$ for $i \in I_u(G)$; determine_module_eval_seq corresponds to OrderedTopologically in line 1 of Algorithm 4; Remaining code corresponds to the traversal of the modules according to this ordering, evaluation of their local computations, and propagation of results from outputs to inputs.

```
def random_specify_hyperparameter(hyperp):                                      1
    assert not hyperp.has_value_assigned()                                      2
                                                                                3
    if isinstance(hyperp, hp.Discrete):                                         4
        v = hyperp.vs[np.random.randint(len(hyperp.vs))]                        5
        hyperp.assign_value(v)                                                  6
    else:                                                                       7
        raise ValueError                                                        8
    return v                                                                    9
                                                                               10
def random_specify(outputs):                                                   11
    hyperp_value_lst = []                                                      12
    for h in co.unassigned_independent_hyperparameter_iterator(outputs):       13
        v = random_specify_hyperparameter(h)                                   14
        hyperp_value_lst.append(v)                                             15
    return hyperp_value_lst                                                    16
                                                                               17
class RandomSearcher(Searcher):                                                18
    def __init__(self, search_space_fn):                                       19
        Searcher.__init__(self, search_space_fn)                               20
                                                                               21
    def sample(self):                                                          22
        inputs, outputs = self.search_space_fn()                               23
        vs = random_specify(outputs)                                           24
        return inputs, outputs, vs, {}                                         25
                                                                               26
    def update(self, val, searcher_eval_token):                               27
        pass                                                                   28
```

Figure 23: Implementation of random search in our language implementation. `sample` assigns values to all the independent hyperparameters in the search space, leading to an architecture that can be evaluated. `update` incorporates the results of evaluating an architecture into the state of the searcher, allowing it to use this information in the next call to `sample`.

the independent hyperparameters of the search space. The role of the search algorithm is to pick values for each of these hyperparameters, leading to a terminal space. Compare to Algorithm 3. `search_space_fn` returns the dictionaries of inputs and outputs for the initial state of the search space (analogous to the search space in frame `a` in Figure 5).

## F    Additional experimental results

We present the full validation and test results for both the search space experiments (Table 3) and the search algorithm experiments (Table 4). For each search space, we performed a grid search over the learning rate with values in $\{0.1, 0.05, 0.025, 0.01, 0.005, 0.001\}$ and an L2 penalty with values in $\{0.0001, 0.0003, 0.0005\}$ for the architecture with the highest validation accuracy Each evaluation in the grid search was trained for 600 epochs with SGD with momentum of $0.9$ and a cosine learning rate schedule We did a similar grid search for each search algorithm.

Table 3: Results for the search space experiments A grid search was performed on the best architecture from the search phase Each evaluation in the grid search was trained for 600 epochs

| Search Space | Validation Accuracy @ 25 epochs | Validation Accuracy @ 600 epochs | Test Accuracy @ 600 epochs | Number of Parameters |
|---|---|---|---|---|
| Genetic [26] | 79.03 | 91.13 | 90.07 | 9.4M |
| Flat [15] | 80.69 | 93.70 | 93.58 | 11.3M |
| Nasbench [27] | 87.66 | 95.08 | 94.59 | 2.6M |
| Nasnet [28] | 82.35 | 94.56 | 93.77 | 4.5M |

Table 4: Results for the search algorithm experiments A grid search was performed on the best architecture from the search phase, each trained to 600 epochs

| Search algorithm | Run | Validation Accuracy @ 25 epochs | Validation Accuracy @ 600 epochs | Test Accuracy @ 600 epochs |
|---|---|---|---|---|
| Random | 1 | 77.58 | 92.61 | 92.38 |
|  | 2 | 79.09 | 91.93 | 91.30 |
|  | 3 | 81.26 | 92.35 | 91.16 |
|  | **Mean** | $79.31 \pm 1.85$ | $92.29 \pm 0.34$ | $91.61 \pm 0.67$ |
| MCTS [29] | 1 | 78.68 | 91.97 | 91.33 |
|  | 2 | 78.65 | 91.59 | 91.47 |
|  | 3 | 78.65 | 92.69 | 91.55 |
|  | **Mean** | $78.66 \pm 0.02$ | $92.08 \pm 0.56$ | $91.45 \pm 0.11$ |
| SMBO [16] | 1 | 77.93 | 93.62 | 92.92 |
|  | 2 | 81.80 | 93.05 | 92.03 |
|  | 3 | 82.73 | 91.89 | 90.86 |
|  | **Mean** | $80.82 \pm 2.54$ | $92.85 \pm 0.88$ | $91.93 \pm 1.03$ |
| Regularized evolution [14] | 1 | 80.99 | 92.06 | 90.80 |
|  | 2 | 81.51 | 92.49 | 91.79 |
|  | 3 | 81.65 | 92.10 | 91.39 |
|  | **Mean** | $81.38 \pm 0.35$ | $92.21 \pm 0.24$ | $91.32 \pm 0.50$ |

## Footnotes

[5]This is often called a thunk in programming languages.