[Reviews · NeurIPS 2019]

Reviewer 1



Disclaimer: I am not in the CV and NLP field and not sure the proposed language can include every thing we want to search in the applications. This work proposed a unified language for search space definition. To the best of my knowledge, this is original work and in my opinion also important as the search space in the NAS field is much more complicated than the normal HPO/BO cases. The overall quality is quite good. However, I have some minor concerns in the search algorithm experiments: Now the "optimize" in Algorithm 4 only pick one from 256 random generated architectures. I agree this is a valid comparison but how many architectures the original search algorithm papers used? How easy it is to implement a new search algorithm in this framework? The paper is written clearly but I think more comments in the Algorithm 1 and 2 can help. It would also be nice to see a comparison of search space definition using the proposed language and the definition in their original paper. I do think this is a significant work on both methodological and empirical side. After reading authors' response ======================== I changed my score to 7.

Reviewer 2



This paper proposes a formal langauge to describe the search space of architecture search problem. This langauge is a domain specific language embedded in python. Users can write modular, composable, and reusable search space by using this langauge. Originality: The contribution is new. This is the first work that tries to provide a formal langauge for the space definition. Quality: The semantics of the langauge is thoroughly described. However, this is an embedded langauge in python. It does not have its own text format. So the syntax of the language is unclear. Because it has to follow the restriction of the host langauge (Python), the grammer of this langauge is also not very concise. This langauge combines the ideas of mordern deep learing frameworks and hyperparameter search frameworks. The idea of hierachycal 'substitution module' has already appears in some deep learning frameworks (e.g. the 'Block" structure in gluon API of mxnet)). Clarity: The paper is well written with adequate background information. Significance: This is a good tool to formulate the search spaces. I expect many people are willing to use it. It will be better if it can support multiple backends (e.g. tensorflow, pytorch, ...) Questions: 1. Besides modularity, how does this language compared to existing ways to specify search space? Can it also reduce the number of lines of code? 2. I don't like the design of A[out].coect(B[]). Deep learning frameworks do not need to explicitly assign these edges in the computational graph. They build the graph by using inputs as arguments and using outputs as return values. 3. How does it handle network transformation based architecture search (e.g. Efficient Architecture Search by Network Transformation, Path-Level Network Transformation for Efficient Architecture Search)? Their search space is basically defined by some network transformation operations. Minors: 1. Add an explanation for Figure 4. Fig. 4 is not cited in the paper.

Reviewer 3



Clarity: very clear overall Originality: original framework to the best of my knowledge Significance: seems an important contribution to the field, this language should facilitate the development of Neural Architecture Search algorithms. Quality: high quality Minor: "While in different settings these input distributions take different forms, our formulation can work with any of them. Next, we summarize some of these alternatives." =>Unclear sentence. what is “these”? section 5.1: I don't know how helpful the introduction of all the formal notations is for the reader, and it might not the most pedagogical way to explain things. Could maybe move this section to supplement and replace it by more extended examples of code like in fig. 1->3, maybe connecting to known frameworks such as TensorFlow or Pytorch?

[Author Response · NeurIPS 2019]

We thank the reviewers for the useful feedback. We would like to reemphasize what we think to be the importance of
our work and answer the main questions raised by the reviewers.

**On the importance of our work** Our programmable and modular language supports architecture search research
and practice by making it easy to encode new search spaces and decoupling the search space and search algorithm
implementations, making it easy to compare different combinations under the same conditions. Search spaces and
search algorithms implemented in our framework can be used by a wide audience in new use-cases[1]. We provide a
well-documented Python implementation of our language.

All reviewers recognized the importance and novelty of our approach: **R1:** "I do think this is a significant work on both
methodological and empirical side." ; "this is original work and in my opinion also important as the search space in
the NAS field is much more complicated than the normal HPO/BO cases." **R2:** "The contribution is new. This is the
first work that tries to provide a formal language for the space definition."; "This is a good tool to formulate the search
spaces. I expect many people are willing to use it." **R3:** "... the authors propose a formal language for encoding search
spaces over arbitrary computational graphs (important contribution). "; "original framework"; "seems an important
contribution to the field, this language should facilitate the development of Neural Architecture Search algorithms."

**Choice of language description [R2, R3]** We describe our language through text and examples for concrete instances
of modules and hyperparameters (Section 4) and through mathematical notation for its components and mechanics
(Section 5). This presentation is a compromise between readability (i.e., concrete examples in our implementation) and
precision (i.e., formal mathematical description). An abstract language to describe concrete examples would be a hurdle
for the reader without being necessarily superior to Python (which is very common in our community). We include
additional information (both through examples in our implementation and formally) in Appendix A.

**Expressivity and simplicity of the language [R1, R2]** We have been able to naturally encode a representative set of
search spaces in the literature with our language constructs. Our language also allows us to represent infinite search
spaces, as substitutions are lazy and can create new hyperparameters and modules. Infinite search spaces are not
possible with current hyperparameter optimization tools. Furthermore, the constructs that we defined allows us to
perform natural variations of the search space easily. Even search spaces defined through local transformations have an
underlying space of reachable architectures. The incremental nature of training can be incorporated in the definition of
the basic modules used, i.e., by keeping track of the weights. We will add additional discussion to the appendix.

**Additions to the final version [R1, R2, R3]** We will clarify the aspects suggested by the reviewers. We will expand
the discussion of Algorithm 1 and 2 to better explain how the traversal functionality supports search algorithms. We
will add a concrete search algorithm implementation to the appendix. We will explain Figure 4 in the context of the
notation introduced in Section 5 (which should clarify the notation). We will expand on the mapping from architectures
in the search space to their deep learning framework implementations. We will include in the appendix one example
with the side-by-side comparison of our language and hyperparameter optimization tools in terms of convenience and
one example on search spaces with infinite architectures.

**Novel heuristics for NAS [R3]** We agree that this is an interesting and very active research topic. Many of these
heuristics will be able to be made available through our language to a wide audience that can experiment with them in
new use-cases. We expect this to have an important effect on the availability of usable NAS implementations.

**Support for multiple backends [R2]** Our current implementation supports multiple backends (Tensorflow, Pytorch,
and Keras). Substitution modules, hyperparameters, and search algorithms are backend independent. It is very easy
to add additional backends, e.g., requires very small and local code changes to basic module helpers. . This enables
applying architecture search to other domains easily (e.g., Sklearn pipelines and data augmentation policies).

**Supporting more search algorithms and search spaces [R1]** Implementing new search algorithms is very easy.
Search algorithms only interface with search spaces through hyperparameter traversal. Algorithms 3 and 4 are illustrative
of search algorithms described through our notation. We will add a concrete example in our implementation to the
appendix. In appendix A, we have additional search space examples, e.g., a more complex search space for the recurrent
cell search space used in ENAS (Figure 8) and some additional discussion on how to implement basic modules (dense
and conv2d in appendix A.3).

**Clarifications for R2 [R2]** **Metrics** We leave the systematic definition of metrics to evaluate search spaces and search
algorithms for future work, as they are better addressed in the context of a NAS benchmark. **Block in Gluon** "Block"
in Gluon is akin to nn.Module in Pytorch. While it allows nesting, contrary to a substitution module, it does not have
architecture search capabilities. **Explicit connects** In the paper, explicit calls to connect are easy to understand and do
not require introducing additional functions. In our implementation, we have other helper functions. **Text format** A text
format representation can be built on top of our implementation. This does not impact the representation capabilities of
our language.

## Footnotes

[1]Architecture will have limited impact without programmable tools, i.e., the ability to easily be used in new problems.


[Meta-Review · NeurIPS 2019]

The authors should be commended for submitting a clear and timely paper on the subject of neural architecture search. The establishment of a formal language describing architectures that allows separation of problem from solution was deemed sufficiently novel to warrant publication. Good work!